# Design, Synthesis, In Vitro, and In Silico Insights of 5-(Substituted benzylidene)-2-phenylthiazol-4(5*H*)-one Derivatives: A Novel Class of Anti-Melanogenic Compounds

**DOI:** 10.3390/molecules28083293

**Published:** 2023-04-07

**Authors:** Dahye Yoon, Min Kyung Kang, Hee Jin Jung, Sultan Ullah, Jieun Lee, Yeongmu Jeong, Sang Gyun Noh, Dongwan Kang, Yujin Park, Pusoon Chun, Hae Young Chung, Hyung Ryong Moon

**Affiliations:** 1Department of Manufacturing Pharmacy, College of Pharmacy, Pusan National University, Busan 46241, Republic of Korea; dahae0528@pusan.ac.kr (D.Y.); kmk87106@pusan.ac.kr (M.K.K.); yijiun@pusan.ac.kr (J.L.); dassabn@pusan.ac.kr (Y.J.); 2Department of Pharmacy, College of Pharmacy, Pusan National University, Busan 46241, Republic of Korea; hjjung2046@pusan.ac.kr (H.J.J.); rskrsk92@pusan.ac.kr (S.G.N.); hyjung@pusan.ac.kr (H.Y.C.); 3Department of Molecular Medicine, The Herbert Wertheim UF Scripps Institute for Biomedical Innovation & Technology, Jupiter, FL 33458, USA; sultanullahf@ufl.edu; 4Department of Medicinal Chemistry, New Drug Development Center, Daegu-Gyeongbuk Medical Innovation Foundation, Daegu 41061, Republic of Korea; kdw4106@kmedihub.re.kr (D.K.); pyj1016@kmedihub.re.kr (Y.P.); 5College of Pharmacy and Inje Institute of Pharmaceutical Sciences and Research, Inje University, Gimhae 50834, Republic of Korea; pusoon@inje.ac.kr

**Keywords:** (*Z*)-BPT derivatives, tyrosinase, melanogenesis, kojic acid, antioxidant, in silico, kinetics

## Abstract

(*Z*)-5-Benzylidene-2-phenylthiazol-4(5*H*)-one ((*Z*)-BPT) derivatives were designed by combining the structural characteristics of two tyrosinase inhibitors. The double-bond geometry of trisubstituted alkenes, (*Z*)-BPTs **1**–**14**, was determined based on the ^3^*J*_C,Hβ_ coupling constant of ^1^H-coupled ^13^C NMR spectra. Three (*Z*)-BPT derivatives (**1**–**3**) showed stronger tyrosinase inhibitory activities than kojic acid; in particular, **2** was to be 189-fold more potent than kojic acid. Kinetic analysis using mushroom tyrosinase indicated that **1** and **2** were competitive inhibitors, whereas **3** was a mixed-type inhibitor. The in silico results revealed that **1**–**3** could strongly bind to the active sites of mushroom and human tyrosinases, supporting the kinetic results. Derivatives **1** and **2** decreased the intracellular melanin contents in a concentration-dependent manner in B16F10 cells, and their anti-melanogenic efficacy exceeded that of kojic acid. The anti-tyrosinase activity of **1** and **2** in B16F10 cells was similar to their anti-melanogenic effects, suggesting that their anti-melanogenic effects were primarily owing to their anti-tyrosinase activity. Western blotting of B16F10 cells revealed that the derivatives **1** and **2** inhibited tyrosinase expression, which partially contributes to their anti-melanogenic ability. Several derivatives, including **2** and **3**, exhibited potent antioxidant activities against ABTS cation radicals, DPPH radicals, ROS, and peroxynitrite. These results suggest that (*Z*)-BPT derivatives **1** and **2** have promising potential as novel anti-melanogenic agents.

## 1. Introduction

Human skin has three layers: the epidermis, dermis, and subcutaneous layer [1]. The epidermis is the top layer of the skin and acts as the first line of defense against external factors, such as pathogens and UV irradiation. The epidermis primarily consists of keratinocytes and melanocytes in the basal layer [1]. Melanocytes contain special organelles called melanosomes, which produce melanin. Melanin produced in the melanosomes of melanocytes is transferred to keratinocytes [2,3]. In mammals, there are two types of melanin: pheomelanin (yellow-to-red pigment) and eumelanin (brown-to-black pigment) [3,4,5]. They are responsible for skin color, which is primarily determined by the amount and ratio of the two types of melanin contained in the keratinocytes of the skin epidermis [6]. Melanin pigments are widely distributed in organisms, including bacteria, fungi, plants, invertebrates, and vertebrates. Melanin has beneficial biological effects, such as UV radiation defense, sunscreen, antioxidant, and free radical and hazardous chemical scavenging [4,7,8,9]. Despite its beneficial effects, the overproduction and abnormal accumulation of melanin in specific areas of the skin cause diverse esthetic problems and hyperpigmentation disorders, including freckles, melasma, senile lentigines, post-inflammatory melanoderma, and skin cancers [2,5,10,11,12]. Although the biosynthetic pathway of melanin in humans is slightly different from that of microorganisms, l-tyrosine is converted to melanin by passing through a series of biochemical reactions involving tyrosinase reactions [13]. Tyrosinase is a biological enzyme that is involved in several biological processes in arthropods, including cuticle hardening, wound healing, and protective encapsulation [14]. In plants, it accelerates the enzymatic browning of damaged crops after harvest or storage [14,15,16,17].

Melanin is produced through a complex process called melanogenesis [18]. Tyrosinase (E.C. 1.14.18.1, polyphenolase) [19,20] is the rate-determining enzyme in melanogenesis [21]. In the production of melanin, tyrosinase participates in three catalytic processes: the conversion of l-tyrosine to l-DOPA via monophenolase activity, the oxidation of l-DOPA to dopaquinone via diphenolase activity, and the conversion of 5,6-dihydroxyindole to indole-5,6-quinone (oxidation by diphenolase activity) [1,21]. The rate-determining steps of melanogenesis, which comprises several complex chemical and enzymatic reactions, are the first and second reactions (l-tyrosine to dopaquinone) [21]. Tyrosinase is regarded as an appealing target for pharmaceutical and therapeutic research on melanogenesis because of its crucial involvement in forming melanin. There are two methods for anti-melanogenesis using tyrosinase inhibition: (i) direct inhibition of tyrosinase activity and (ii) inhibition of tyrosinase expression [22]. Since the microphthalmia-associated transcription factor (MITF) regulates the gene expression of the enzymes involved in melanin biosynthesis, such as tyrosinase-related protein 1 (TRP-1), TRP-2, and tyrosinase [23], it is considered a key regulator of melanin biosynthesis. The direct inhibition of tyrosinase activity, along with the modulation of tyrosinase expression, is one of the most appealing methods to regulate melanogenesis. Tyrosinase inhibitors are widely studied for their role in the development of cosmetic and therapeutic skin-whitening agents and agricultural pesticides [24,25]. A great number of tyrosinase inhibitors of natural, synthetic, and semi-synthetic origins have been identified; however, only a few are being used for the treatment of disorders that are associated with hyperpigmentation owing to insufficient clinical efficacy and adverse effects [26,27,28,29]. Therefore, there remains a considerable demand for novel tyrosinase inhibitors that are more effective and safer than the tyrosinase inhibitors identified to date.

In previous studies over the past 15 years, we reported that various compounds bearing a β-phenyl-α,β-unsaturated carbonyl (PUSC) template had potent tyrosinase inhibitory activities in vitro (mushroom and murine B16F10 cells) and in vivo [30,31,32,33,34,35,36,37,38,39]. As shown in Figure 1, (*Z*)-5-benzylidenethiazolidine-2,4-dione (**I**) [40] and (*E*)-2-benzylidene-2,3-dihydro-1*H*-inden-1-one (**II**) [41] derivatives with a PUSC template also exhibited excellent tyrosinase inhibition and anti-melanogenic efficacy. In particular, (*Z*)-5-(2,4-dihydroxybenzylidene)thiazolidine-2,4-dione, one of the former derivatives, is used as an ingredient in skin-whitening agents in South Korea. A novel class of tyrosinase inhibitors was designed based on the structural characteristics of both the compounds. As shown in Figure 1, the thiazolone ring of the target compounds was introduced by mimicking the thiazolidinedione of compound **I**. In addition, a benzene ring was introduced into the thiazolone ring of the target compound to mimic the benzene ring fused to the 5-membered ring of compound **II**. Since melanocytes are present in the bottom layer of the epidermis, the effect of tyrosinase inhibitors on anti-melanogenesis may be related to their skin absorbance. Log *P* (partition coefficient) values [42] indicate how hydrophobic or hydrophilic a compound is, and the higher the log *P* value, the more hydrophobic the compound is. Therefore, compounds with higher log *P* values are more likely to be absorbed by the skin. As indicated in Figure 1, according to the results obtained using ChemDraw Ultra 12.0, the log *P* values of the benzylidene compounds on thiazolidinedione, indenone, and 2-phenylthiazolone scaffolds were 1.49, 3.51, and 3.94, respectively. These results encouraged us to synthesize the target compounds, (*Z*)-5-benzylidene-2-phenylthiazol-4(5*H*)-ones ((*Z*)-BPTs), as potential anti-melanogenic compounds.

## 2. Results and Discussion

### 2.1. Synthesis of (Z)-BPT Derivatives ***1***–***14***

As described in Figure 1, the key intermediate **16** [43] for synthesizing (*Z*)-BPT derivatives **1**–**14** was prepared via a two-step reaction: i) cyclization and ii) HBr salt removal. The reflux of thiobenzamide and bromoacetic acid in ethyl acetate produced 2-phenylthiazol-4(5*H*)-one hydrobromide **15** with a 69% yield, and treatment with pyridine produced 2-phenylthiazol-4(5*H*)-one **16** with a 92% yield. The condensation of **16** with diverse substituted benzaldehydes (**a**–**n**) in the presence of piperidine (0.3 equiv.) in refluxed ethyl alcohol generated the desired (*Z*)-BPT derivatives **1**–**14** as single products with yields of 41–81%. The structures of the final compounds were confirmed using NMR (^1^H and ^13^C) spectroscopy and mass spectrometry (Appendix A). The double-bond stereochemistry of the trisubstituted alkenes **1**–**14**, generated from the condensation between benzaldehydes and **16**, was determined based on the ^3^*J* (between vicinal ^1^H and ^13^C) coupling constants of ^1^H-coupled ^13^C NMR spectra. Nair et al. previously reported that the double-bond geometries in trisubstituted enamides could be differentiated based on the observed vicinal ^1^H and ^13^C-coupling constant (^3^*J*_C,Hβ_) of the amide carbonyl carbon [44]. According to the report, vicinal coupling constants ranging from 3.6 to 7.0 Hz indicate that H_β_ and the carbonyl carbon of the amide are on the same side. On the other hand, the vicinal coupling constants (≥7.5 Hz), which are usually greater than 10 Hz, indicate that H_β_ and the carbonyl carbon are on the opposite sides. To confirm the double-bond geometry of the (*Z*)-BPT derivatives, the ^13^C NMR of **2** was measured in the ^1^H-coupled NMR mode. The amide carbonyl carbon had a vicinal ^1^H and ^13^C-coupling constant of 5.3 Hz (see S6. The proton-coupled ^13^C NMR spectrum of compound **2** in the Appendix A), suggesting that **2** has a (*Z*)-geometry.

### 2.2. Tyrosinase Inhibition of (Z)-BPT Derivatives

Although melanogenesis is modulated by hormones, tyrosinase inhibition is the most effective method for controlling melanogenesis. Therefore, the inhibitory effect of 14 synthetic (*Z*)-BPT derivatives on mushroom tyrosinase was investigated according to a previously described procedure [39]. For evaluating tyrosinase inhibitors, l-tyrosine and kojic acid were utilized as a substrate and a positive control, respectively. All the 14 (*Z*)-BPT derivatives inhibited the mushroom tyrosinase activity in a concentration-dependent manner. Table 1 shows the IC_50_ values of (*Z*)-BPT derivatives and kojic acid.

The (*Z*)-BPT derivatives **8**–**11**, with no hydroxyl group, showed no potent tyrosinase inhibition (IC_50_ > 300 µM). Although derivative **7** contained a 2-hydroxyl substituent on the β-phenyl ring of the PUSC template, it did not exhibit mushroom tyrosinase inhibitory activity. Derivative **1** contained a 4-hydroxyl substituent on the β-phenyl ring and exhibited a strong tyrosinase inhibitory activity (IC_50_ = 6.4 ± 0.52 µM), and inhibited mushroom tyrosinase more strongly than kojic acid (IC_50_ = 20.8 ± 1.34 µM). The introduction of a 3-bromo substituent on the β-phenyl ring of derivative **1** dramatically reduced the tyrosinase inhibitory activity (IC_50_ value of derivative **13**: 253.7 ± 1.87 μM). However, when an additional 5-bromo substituent was inserted into the β-phenyl ring of derivative **13**, the tyrosinase inhibitory activity increased to some extent (IC_50_ value of derivative **14**: 87.2 ± 0.63 µM). The introduction of a 3-alkoxyl substituent on the β-phenyl ring of derivative **1** also decreased the tyrosinase inhibitory activity (IC_50_ values of derivative **4** with a 3-methoxyl and derivative **5** with a 3-ethoxyl: 211.1 ± 1.84, and 73.8 ± 2.02 µM, respectively). The exchange of 3-methoxyl and 4-hydroxyl substituents in derivative **4** slightly increased the tyrosinase inhibitory activity (IC_50_ value of derivative **6**: 121.1 ± 2.72 µM). The introduction of an additional 3-hydroxyl substituent on the β-phenyl ring of derivative **1** did not influence the tyrosinase inhibitory activity, whereas the insertion of an additional 2-hydroxyl substituent greatly enhanced the inhibition of tyrosinase activity. Derivative **2** contained a 2,4-dihydroxyl substituent and had an IC_50_ value of 0.1 ± 0.01 µM, indicating that **2** was a 190-fold stronger inhibitor than kojic acid. These findings imply that the 4-hydroxyl substituent on the β-phenyl ring of the PUSC template in (*Z*)-BPT derivatives plays a significant role in inhibiting tyrosinase activity, and that the type and position of additional substituents are closely correlated with the level of tyrosinase activity inhibition. As in previous studies [17,45], derivative **2** with 2,4-dihydroxyphenyl exhibited the strongest tyrosinase inhibition.

Log *P* represents the logarithm of the partition coefficient (*P*). The greater the log *P* value of a compound, the better its absorption by the skin. The log *P* values of (*Z*)-BPT derivatives and kojic acid were obtained from ChemDraw Ultra 12.0. As shown in Table 1, kojic acid had a log *P* value of −2.45, whereas (*Z*)-BPT derivatives had log *P* values ranging from 3.16 to 4.38.

Tyrosinase inhibition experiments were performed independently in triplicates. l-tyrosine and kojic acid (KA) were used as the substrate and positive controls, respectively. The IC_50_ values are presented as the mean ± standard error of the mean (SEM). ChemDraw Ultra 12.0 was used to obtain the log *P* values.

### 2.3. Identifying the Inhibition Type of (Z)-BPT Derivatives ***1***–***3***

Since (*Z*)-BPT derivatives **1**–**3** exhibited more potent tyrosinase inhibitory activity than kojic acid, further studies were performed to determine their inhibition type. To investigate the type of inhibition of mushroom tyrosinase, kinetic studies of **1**–**3** were conducted using mushroom tyrosinase with various concentrations of l-DOPA as a substrate. As depicted in Figure 2, the mechanisms of tyrosinase inhibition were analyzed using Lineweaver–Burk plots. (*Z*)-BPT derivatives **1** and **2** showed similar Lineweaver–Burk plot patterns. Each of the four straight lines produced at each inhibitor concentration converged to a single point on the y-axis. Each V_max_ value was constant, regardless of the inhibitor (**1** and **2**) concentration, and each K_M_ value gradually increased as the concentration of each inhibitor increased. These results imply that the (*Z*)-BPT derivatives **1** and **2** bind to the same binding pocket as l-DOPA, a tyrosinase substrate, and inhibit mushroom tyrosinase activity in a competitive manner (Table 2). In contrast, in derivative **3**, four straight lines generated at each inhibitor concentration intersected at a point in the second quadrant. According to the Lineweaver–Burk plot, as the inhibitor concentration increased, the V_max_ value of **3** decreased, but the K_M_ value increased. These results suggest that (*Z*)-BPT derivative **3** is a mixed-type inhibitor.

To determine the K_i_ (inhibition constant) values of (*Z*)-BPT derivatives **1**–**3**, Lineweaver–Burk plots for **1**–**3** were transformed into the corresponding Dixon plots (Figure 3). Each Dixon plot obtained from each Lineweaver–Burk plot produced four straight lines that intersected at one point in the second quadrant. The vertical lines from the merge point to the x-axis provide the K_i_ value of each (*Z*)-BPT derivative. The K_i_ values of (*Z*)-BPT derivatives **1**–**3** were 8.89, 0.10, and 4.80 μM, respectively, indicating that **2** generates the strongest enzyme-inhibitor complexes.

### 2.4. In Silico Docking of (Z)-BPT Derivatives ***1***–***3*** with Mushroom Tyrosinase

Using (*Z*)-BPT derivatives **1**–**3**, which exhibit effective tyrosinase inhibition against mushroom tyrosinase, an in silico docking simulation was performed to examine their behavior in the mushroom tyrosinase active site. Schrodinger Suite (release 2021-2) and kojic acid were used as the docking software and a positive reference, respectively. The results of the in silico docking simulations are shown in Figure 4.

As in kojic acid, all the three (*Z*)-BPT derivatives **1**–**3** bound to the active site of tyrosinase and showed similar or greater binding affinities than kojic acid with a docking score of −4.5 kcal/mol (**1**: −4.3, **2**: −6.2, and **3**: −6.3 kcal/mol) (Figure 4B). Interestingly, all compounds (kojic acid and **1**–**3**) interacted with one or two copper ions in the active site. Kojic acid was found to interact with tyrosinase through three modes: metal coordination between Cu401 and a 2-hydroxymethyl, a hydrogen bond between a 5-hydroxyl and Met280, and π-π stacking between Ser282 and a pyranone ring. In (*Z*)-BPT derivative **1**, a metal coordination between a phenolic 4-hydroxyl and Cu401 was formed, and three π–π stacking interactions (between a β- phenyl ring and His259 and His263, and a 2-phenyl ring and His244) were created. Unlike (*Z*)-BPT derivative **1**, the phenolic 4-hydroxyl of (*Z*)-BPT derivative **2** interacted with two copper ions (Cu400 and Cu401), not a single copper ion, through a salt bridge rather than a metal coordination. In addition, as seen in **1**, **2** also formed three π-π stacking interactions using the same interaction patterns as **1**. Furthermore, the 2-hydroxyl group on the β-phenyl ring of **2** participated in hydrogen bonding interactions with Glu256. The phenolic 4-hydroxyl of derivative **3** generated two salt bridges with Cu400 and Cu401, as in **2**, and the 2-phenyl ring formed one π-cation interaction with Arg268. These results suggest that (*Z*)-BPT derivatives **1**–**3** can bind to the active site of tyrosinase via various interactions, including interactions with copper ions, such as kojic acid.

### 2.5. In Silico Docking of (Z)-BPT Derivatives ***1***–***3*** with the Human Tyrosinase Model

Since the X-ray crystal structure of human tyrosinase is not available, we previously designed a human tyrosinase (*h*TYR) homology model based on tyrosinase-related protein-1. To predict the binding ability to human tyrosinase, (*Z*)-BPT derivatives **1**–**3** were docked with a previously designed *h*TYR homology model [34], and kojic acid was used as a reference material. The docking simulation results are shown in Figure 5.

Kojic acid interacted with Zn7, Met374, and Ser375 through metal coordination, π–π stacking, and hydrogen bonding, respectively (Figure 5A). These interactions provided a docking score of −4.4 kcal/mole (Figure 5B). Derivative **1** participated in two interactions (Figure 5C). The 4-hydroxyl of the β-phenyl ring in **1** hydrogen-bonded with Arg196, and the 2-phenyl ring interacted with Phe347 via π-π stacking. The docking score was −4.1 kcal/mol. Derivative **2** showed similar interactions with mushroom tyrosinase (Figure 5D). The 4-hydroxyl of the β–phenyl ring in **2** participated in the formation of salt bridges with two zinc ions (Zn6 and Zn7), the 2-hydroxyl of the β–phenyl ring generated a hydrogen bond with Met374, and the β–phenyl ring produced a π-π stacking interaction with His367. These interactions with the homology model afforded a high binding affinity with a docking score of −7.2 kcal/mol. In contrast, although derivative **3** had two hydroxyl groups on the β-phenyl ring like **2**, the hydroxyls of **3** did not interact with zinc ions (Figure 5E). Instead, the two hydroxyl groups formed two hydrogen bonds with Ser184 and Arg196. In addition, the 2-phenyl ring participated in π-π stacking with Phe347. These interactions provided a slightly lower docking score (−3.9 kcal/mol) than that of kojic acid. These results suggest that the (*Z*)-BPT derivative **2** has potent anti-*h*TYR activity.

### 2.6. Effects of (Z)-BPT Derivatives ***1***–***3*** on B16F10 Cell Viability

To examine whether (*Z*)-BPT derivatives **1**–**3** had the ability to exhibit inhibitory effects on melanogenesis and cellular tyrosinase activity, B16F10 murine melanoma cells were utilized for cell experiments. Before investigating their inhibitory effects on B16F10 cells, the viability of (*Z*)-BPT derivatives **1**–**3** was examined in B16F10 cells. Cell viability was monitored for 72 h using the EZ-Cytox assay, and the results are shown in Figure 6.

To determine the cell viability, six different concentrations (0, 1, 2, 5, 10, and 20 µM) of (*Z*)-BPT derivatives **1**–**3** were used, and the results were examined at two time points (48 and 72 h). Derivatives **1** and **2** did not show any perceptible cytotoxicity at concentrations ≤ 20 µM at either time point, but derivative **3** showed cytotoxic effects at concentrations ≥ 5 µM at 72 h and at concentrations ≥ 2 µM at 72 h. Based on the cell viability results, the concentrations for cell-based assays related to the melanin production level and cellular tyrosinase activity were determined.

### 2.7. Effects of (Z)-BPT Derivatives ***1*** and ***2*** on Melanin Production in B16F10 Cells

Derivative **3** showed an excellent inhibitory activity against mushroom tyrosinase but exhibited cytotoxicity in B16F10 cells. Thus, the experiments evaluating the effect of **3** on cellular tyrosinase activity and melanin production in B16F10 cells were excluded. To confirm whether (*Z*)-BPT derivatives **1** and **2**, which show potent mushroom tyrosinase inhibitory activity, can exert anti-melanogenic effects in mammalian cells, the melanin levels were measured in B16F10 cells. Cells were exposed to four different concentrations (0, 5, 10, and 20 µM) of **1** and **2** prior to treatment with stimulators (3-isobutyl-1-methylxanthine (IBMX, 200 µM), and α-melanocyte stimulating hormone (α-MSH, 1 µM)). After 72 h of incubation, the effects of **1** and **2** on melanin production were determined. Cellular melanin levels were assessed by measuring melanin levels in cell lysates. The results are shown in Figure 7.

B16F10 cells treated with stimulators (IBMX and α-MSH) greatly increased the cellular melanin content by 3.3-fold compared to the untreated control (100%). However, exposure to **1** and **2** significantly and dose-dependently diminished the cellular melanin content that was increased by the stimulators. Derivative **1** at 20 µM reduced the cellular melanin content more effectively than kojic acid at the same concentration. However, derivative **2** exhibited more remarkable results. The inhibitory effect of **2** at 5 µM on the cellular melanin content was similar to that of 20 µM kojic acid, and derivative **2** at 20 µM reduced the cellular melanin levels that were enhanced by stimulators to levels below those of the non-treated control.

### 2.8. Effects of (Z)-BPT Derivatives ***1*** and ***2*** on B16F10 Cellular Tyrosinase Activity

To examine the origin of the anti-melanogenic effects of (*Z*)-BPT derivatives **1** and **2**, the anti-tyrosinase effects of these derivatives were investigated using B16F10 cells. Similar to the melanin production assay, four different (*Z*)-BPT derivative concentrations (0, 5, 10, and 20 µM of **1** and **2**) were exposed to B16F10 cells prior to treatment with stimulators, IBMX (200 µM) and α-MSH (1 µM). After 72 h of incubation, the effects of **1** and **2** on cellular tyrosinase activity were analyzed. The results are depicted in Figure 8; kojic acid was used as the positive control.

Treatment of stimulators (IBMX and α-MSH) with B16F10 cells highly increased the cellular tyrosinase activity by 3.3-fold compared to that of the untreated control (100%). Kojic acid (20 µM) treatment diminished the enhanced cellular tyrosinase activity by 2.4-fold. Treatment with derivatives **1** and **2** dose-dependently and significantly decreased the cellular tyrosinase activity, which was enhanced by treatment with stimulators. Derivative **1** exerted a stronger inhibitory activity than kojic acid at 20 µM. Derivative **2** had a more potent cellular anti-tyrosinase activity than derivative **1** and kojic acid. At a concentration of 5 µM, derivative **2** had cellular tyrosinase inhibitory activity comparable to that of 20 µM kojic acid, and at a concentration of 20 µM, it reduced the cellular tyrosinase activity to the levels observed in the untreated control. The results of derivatives **1** and **2** for the cellular tyrosinase activity were similar to those for cellular melanin content, implying that the anti-melanogenic capacity of **1** and **2** in B16F10 cells was mainly attributable to their ability to inhibit cellular tyrosinase.

### 2.9. ABTS Cation Radical Scavenging Effects of (Z)-BPT Derivatives ***1***–***14***

It has been suggested that substances with antioxidant qualities may prevent the tyrosinase substrate (l-tyrosine and l-dopa) oxidation required for melanogenesis without interacting with the enzyme [46,47]. In addition, it has been reported that antioxidants that scavenge radicals may inhibit excess melanogenesis, because melanogenesis can be triggered by various radicals [48,49,50,51,52]. Therefore, the 2,2′-azino-bis(3-ethylbenzothiazoline-6-sulfonic acid) (ABTS) cation radical scavenging abilities of 14 (*Z*)-BPT derivatives were studied using Trolox as a positive control. The ABTS cation radical is formed by transferring one electron of ABTS to K_2_S_2_O_8_ (potassium persulfate). To obtain an ABTS cation radical solution, a 7.4 mM ABTS solution was mixed with 2.6 mM potassium persulfate and kept in the dark at room temperature for 24 h. After adjusting the absorbance of the ABTS cation radical solution to 0.700 ± 0.03 at 732 nm, the ABTS cation radical solution was mixed with test samples (**1**–**14** and Trolox) to a final concentration of 100 µM and kept for 2 min in the dark. The radical scavenging effects were assessed based on the absorbance measured at 732 nm. Figure 9 shows the ABTS radical scavenging activity.

Of the fourteen (*Z*)-BPT derivatives, four derivatives, **2**–**5**, with a 2,4-dihydroxyl, 3,4-dihydroxyl, 4-hydroxy-3-methoxyl, and 3-ethoxy-4-hydroxyl on the β-phenyl ring, respectively, showed strong ABTS cation radical scavenging abilities with a range of 56–67% inhibition. On the other hand, Trolox exhibited the strongest radical scavenging effect (81% inhibition). Three derivatives, **6**, **7**, and **12**, were moderate radical scavengers with a range of 34–43% inhibition, and the remaining derivatives showed weak radical scavenging effects with less than 23% inhibition. Derivatives **2** and **3**, with two hydroxyls on the β-phenyl ring, exerted stronger ABTS cation radical scavenging abilities than derivatives with one or no hydroxyl on the β-phenyl ring. Derivatives **4**–**6**, with one alkoxyl substituent and one hydroxyl substituent at positions three and four of the β-phenyl ring, showed moderate to strong ABTS^+^ radical scavenging abilities, and derivatives **4** and **5**, with a 3-alkoxy-4-hydroxyl substituent, exhibited more potent radical scavenging activities than those of derivative **6**, with a 3-hydroxy-4-alkoxyl substituent.

### 2.10. DPPH Radical Scavenging Effects of (Z)-BPT Derivatives ***1***–***14***

The 2,2-diphenyl-1-picrylhydrazyl (DPPH) radical scavenging capacity of the (*Z*)-BPT derivatives (1 mM) was investigated using l-ascorbic acid (1 mM, vitamin C) as a positive control. Figure 10 shows the radical scavenging outcomes. A total of 7 of the 14 (*Z*)-BPT derivatives (1–5, 12, and 14) showed moderate to strong DPPH radical scavenging activity compared with l-ascorbic acid. While (*Z*)-BPT derivatives 8–11, with no hydroxyl substituents on the β-phenyl ring of the PUSC template, did not demonstrate any appreciable DPPH radical scavenging activity, those with hydroxyl substituents typically demonstrated moderate to strong radical scavenging activity. l-Ascorbic acid scavenged 97.1% of the DPPH radical, whereas (*Z*)-BPT derivative 1 with a 4-hydroxyl substituent on the β-phenyl ring scavenged 46.5% of the DPPH radical. Interestingly, when a 3-bromo substituent was inserted into the β-phenyl ring of 1, the radical scavenging activity decreased to 14.8% (derivative 13), but when two bromo substituents were introduced at positions three and five of the β-phenyl ring of 1, the radical scavenging activity increased to 57.3% (derivative 14). The strongest radical scavenging activity was observed in derivative 3 with a β-3,4-dihydroxyphenyl moiety, which scavenged 96.2% of the DPPH radical, similar to l-ascorbic acid. Derivative 4, with a 4-hydroxy-3-methoxyl substituent, showed a strong DPPH radical scavenging activity of 81.0%, whereas derivative 6, in which the positions of the hydroxyl group and the methoxy groups on the β-phenyl ring of 4 were exchanged, showed a low radical scavenging activity of only 25.1%. Derivative 2, with a 2,4-dihydroxylphenyl ring, and derivative 5, with a 3-ethoxy-4-hydroxylphenyl ring, also showed good DPPH radical scavenging activities of 66.9 and 58.0%, respectively.

### 2.11. ROS Scavenging Effects of (Z)-BPT Derivatives ***1***–***14***

Owing to the association between antioxidant activity and anti-melanogenesis [46,47,53], the ROS scavenging effect of (*Z*)-BPT derivatives was evaluated. To assess the ROS scavenging activities of the (*Z*)-BPT derivatives, two assays were used to measure intracellular ROS and in vitro ROS generated by 3-morpholinosydnonimine (SIN-1).

The intracellular ROS scavenging activity was measured using the assay method reported by Bondy and Lebel [54] and Ali et al. [55]. We used 2′,7′-dichlorodihydrofluorescein diacetate (DCFH-DA) to measure intracellular ROS. DCFH-DA is converted to 2′,7′-dichlorodihydrofluorescein (DCFH), a hydrolyzed form of DCFH-DA, which in turn, reacts with intracellular ROS to generate 2′,7′-dichlorofluorescein (DCF), a fluorescent material. B16F10 cells treated with 10 µM 3-morpholinosydnonimine (SIN-1), an ROS/RNS generator, greatly enhanced intracellular ROS levels (Figure 11A). Of the fourteen (*Z*)-BPT derivatives, nine derivatives (**1**–**6** and **12**–**14**) at a concentration of 20 µM significantly decreased the intracellular ROS levels enhanced by SIN-1 treatment. In particular, five derivatives (**3**–**5**, **12** and **14**) showed similar or more potent intracellular ROS scavenging activities than Trolox, a well-known antioxidant. These compounds have 3,4-dihydroxyl (catechol) (**3**), 3-alkoxy-4-hydroxyl (**4** and **5**), 3,5-dimethyl-4-hydroxyl (**12**), and 3,5-dibromo-4-hydroxyl (**14**) substituents on the β-phenyl ring, respectively.

DCFH-DA, esterase, and SIN-1 were used to measure the scavenging activity of in vitro ROS generated by SIN-1. DCFH-DA reacts with esterase to generate DCFH, which in turn, reacts with the in vitro ROS generated by SIN-1 to produce DCF. To evaluate the in vitro ROS scavenging effects of the (*Z*)-BPT derivatives, a concentration of 40 µM was used. SIN-1 treatment greatly increased the ROS levels in vitro, and treatment with Trolox, a positive control, significantly decreased the enhanced ROS levels (Figure 11B). As observed for Trolox, five (*Z*)-BPT derivatives (**2**–**4**, **13**, and **14**) also significantly reduced the ROS levels that were increased by SIN-1 treatment to a similar or greater extent than did Trolox. Three derivatives, **2**, **3**, and **14**, exerted strong scavenging abilities against both intracellular ROS and in vitro ROS. In particular, derivatives **2** and **3**, which showed potent inhibitory activities against both tyrosinases in mushroom and B16F10 cells, had the most potent ROS scavenging activity.

### 2.12. Peroxynitrite (ONOO^−^) Scavenging Effects of (Z)-BPT Derivatives ***1***–***14***

Reactive nitrogen species (RNS) have been reported to be involved in melanogenesis induction [56]. Thus, the peroxynitrite scavenging effects of the (*Z*)-BPT derivatives were examined using SIN-1, an RNS/ROS generator, and dihydrorhodamine (DHR123), an ROS indicator. SIN-1 generates peroxynitrite, which, in turn, oxidizes DHR123 to produce rhodamine 123, a fluorescent material. Therefore, the peroxynitrite scavenging activities of the (*Z*)-BPT derivatives **1***–***14** were determined at 50 µM by measuring the fluorescence of rhodamine 123. The peroxynitrite scavenging activity of the (*Z*)-BPT derivatives is shown in Figure 12.

Of the 14 derivatives, 13 significantly reduced peroxynitrite levels that were enhanced by treatment with SIN-1. Notably, eight derivatives (**1**–**5** and **12**–**14**), with a 4-hydroxyl substituent on the β-phenyl ring, showed potent peroxynitrite scavenging efficacy and were stronger peroxynitrite scavengers than penicillamine, which was used as the positive control. These results indicated that derivatives **2** and **3** are potent tyrosinase inhibitors with strong antioxidant capacities.

### 2.13. Effects of (Z)-BPT Derivatives ***1***–***2*** on Tyrosinase Expression

We investigated whether the derivatives **1** and **2** could influence the expression of tyrosinase protein using Western blotting. As shown in Figure 13, the tyrosinase expression levels in B16F10 cells were elevated via stimulator treatment (200 µM of IBMX plus 1 µM of α-MSH) by 6.6-fold for **1** and 5.7-fold for **2**, compared to those in the untreated control. Treatment with derivatives **1** and **2** significantly reduced the tyrosinase expression levels in a dose-dependent manner. At 20 µM, derivatives **1** and **2** reduced the tyrosinase expression levels that increased during the stimulator treatment to 4.4- and 3.1-fold, respectively, implying that derivatives **1** and **2** may, in part, contribute to their anti-melanogenic actions by inhibiting tyrosinase expression.

## 3. Materials and Methods

### 3.1. Reagents

The chemicals 4-(1,1,3,3-tetramethylbutyl)phenylpolyethylene glycol (Triton^TM^ X-100), l-4-hydroxyphenylalanine (l-tyrosine), dimethyl sulfoxide (DMSO), kojic acid, 3-isobutyl-1-methylxanthine (IBMX), potassium hydrogen phosphate, mushroom tyrosinase, potassium dihydrogen phosphate, alpha-melanocyte-stimulating hormone (α-MSH), l-3,4-dihydroxyphenylalanine (l-dopa), and phenylmethylsulfonyl fluoride (PMSF) were purchased from Sigma-Aldrich (St. Louis, MO, USA).

### 3.2. Chemistry

#### 3.2.1. General Methods

Thermo Fisher Scientific (Carlsbad, CA, USA), Sigma-Aldrich (St. Louis, MO, USA), and SEJIN CI Co. (Seoul, Republic of Korea) provided the chemical reagents, which were all used without additional purification. All solvents requiring anhydrous conditions were distilled over CaH_2_ or Na/BP. All reactions were carried out under nitrogen, and the progress of the reactions were monitored via thin-layer chromatography (TLC) using Merck pre-coated 60F_245_ plates. Reaction mixture was purified via flash column chromatography using MP Silica 40-63 (60 Å). Low-resolution mass data were measured using the ESI positive or negative mode on an Expression CMS mass spectrometer (Advion, Ithaca, NY, USA). A Varian Unity AS500 unit (Agilent technologies, Santa Clara, CA, USA) was used to record the ^1^H-NMR spectra at 500 MHz, and a Varian Unity INOVA 400 instrument (Agilent technologies, Santa Clara, CA, USA) was used to record the ^1^H and ^13^C-NMR spectra at 400 MHz and 100 MHz, respectively. Chloroform-*d* and dimethylsulfoxide-*d*_6_ were used as the NMR solvents. Parts per million (ppm) measurements of all chemical shifts were made in comparison to the corresponding residual solvent or deuterated peaks (*δ*_C_ 39.8 and *δ*_H_ 2.48 for DMSO-*d*_6_; *δ*_C_ 77.0 and *δ*_H_ 7.27 for CDCl_3_). The values for the coupling constant (*J*) are presented in hertz (Hz). For ^1^H NMR, the following abbreviations were used: m, multiplet; brs, broad singlet; s, singlet; d, doublet; dd, doublet of doublets; t, triplet; and q, quartet.

#### 3.2.2. General Procedure for the Synthesis of (Z)-BPT Derivatives **1**–**14**

A solution of **16** (100 mg, 0.56 mmol) and an appropriate benzaldehyde (1.0 equiv.) in ethanol (3 mL) was refluxed in the presence of piperidine (0.02 mL, 0.17 mmol) as a base catalyst for 0.5–2.5 h. After cooling, water was added to the reaction mixture. The resultant solid was filtered and washed with water and ethanol for compounds **1**, **2**, **4**–**8**, and **11**–**14**, or with ethyl acetate and hexane (1:1) for compound **3**. For compounds **9** and **10**, the filter cake was further purified through silica gel column chromatography using chloroform and methanol (70:1) for compound **9**, or methylene chloride and ethyl acetate (20:1) for compound **10**.

Compound **1**: (*Z*)-5-(4-Hydroxybenzylidene)-2-phenylthiazol-4(5*H*)-one.

Yield: 81%; ^1^H NMR (500 MHz, DMSO-*d*_6_) *δ* 8.15 (d, 2H, *J* = 8.0 Hz), 7.90 (s, 1H), 7.75 (t, 1H, *J* = 6.5 Hz), 7.65–7.60 (m, 4H), and 6.86 (d, 2H, *J* = 8.0 Hz); ^13^C NMR (100 MHz, DMSO-*d*_6_) *δ* 185.8, 183.0, 163.6, 139.6, 135.8, 134.2, 132.1, 130.3, 128.9, 124.0, 121.4, and 117.8; and LRMS (ESI–) *m/z* 280 (M–H)^−^.

Compound **2**: (*Z*)-5-(2,4-Dihydroxybenzylidene)-2-phenylthiazol-4(5*H*)-one.

Yield: 76%; ^1^H NMR (400 MHz, DMSO-*d*_6_) *δ* 10.63 (brs, 2H), 8.25 (s, 1H), 8.15 (d, 2H, *J* = 8.0 Hz), 7.76 (t, 1H, *J* = 8.0 Hz), 7.63 (t, 2H, *J* = 8.0 Hz), 7.49 (d, 1H, *J* = 8.4 Hz), 6.45 (dd, 1H, *J* = 8.4, 2.0 Hz), and 6.44 (d, 1H, *J* = 2.0 Hz); ^13^C NMR (100 MHz, DMSO-*d*_6_) *δ* 185.6, 183.2, 163.7, 161.3, 135.7, 134.2, 132.2, 131.4, 130.3, 128.9, 120.4, 113.0, 109.7, and 103.2; and LRMS (ESI–) *m/z* 296 (M–H)^−^.

Compound **3**: (*Z*)-5-(3,4-Dihydroxybenzylidene)-2-phenylthiazol-4(5*H*)-one.

Yield: 72%; ^1^H NMR (400 MHz, DMSO-*d*_6_) *δ* 10.11 (s, 1H), 9.54 (s, 1H), 8.16 (d, 2H, *J* = 8.0 Hz), 7.86 (s, 1H), 7.79 (t, 1H, *J* = 8.0 Hz), 7.66 (t, 2H, *J* = 8.0 Hz), 7.23 (d, 1H, *J* = 2.4 Hz), 7.19 (dd, 1H, *J* = 8.5, 2.5 Hz), and 6.91 (d, 1H, *J* = 8.5 Hz); ^13^C NMR (100 MHz, DMSO-*d*_6_) *δ* 185.9, 182.7, 150.4, 146.5, 139.6, 135.7, 131.8, 130.1, 128.7, 125.6, 1258.2, 122.0, 117.4, and 116.8; and LRMS (ESI–) *m/z* 296 (M–H)^−^.

Compound **4**: (*Z*)-5-(4-Hydroxy-3-methoxybenzylidene)-2-phenylthiazol-4(5*H*)-one.

Yield: 72%; ^1^H NMR (400 MHz, CDCl_3_) *δ* 8.20 (d, 2H, *J* = 7.6 Hz), 7.98 (s, 1H), 7.68 (t, 1H, *J* = 7.6 Hz), 7.49 (t, 2H, *J* = 7.6 Hz), 7.27 (dd, 1H, *J* = 8.4, 2.0 Hz), 7.13 (d, 1H, *J* = 2.0 Hz), 7.04 (d, 1H, *J* = 8.4 Hz), and 3.99 (s, 3H); ^13^C NMR (100 MHz, CDCl_3_) *δ* 186.7, 183.7, 149.2, 147.2, 139.4, 135.2, 132.1, 129.4, 129.0, 126.6, 126.0, 123.7, 115.6, 112.9, and 56.3; and LRMS (ESI–) *m/z* 310 (M–H)^−^.

Compound **5**: (*Z*)-5-(3-Ethoxy-4-hydroxybenzylidene)-2-phenylthiazol-4(5*H*)-one.

Yield: 54%; ^1^H NMR (400 MHz, CDCl_3_) *δ* 8.19 (d, 2H, *J* = 8.0 Hz), 7.97 (s, 1H), 7.67 (t, 1H, *J* = 8.0 Hz), 7.55 (t, 2H, *J* = 8.0 Hz), 7.27 (d, 1H, *J* = 8.4 Hz), 7.12 (s, 1H), 7.04 (d, 1H, *J* = 8.4 Hz), 6.16 (s, 1H), 4.21 (q, 2H, *J* = 6.8 Hz), and 1.52 (t, 3H, *J* = 6.8 Hz); ^13^C NMR (100 MHz, CDCl_3_) *δ* 186.7, 183.7, 149.2, 146.5, 139.5, 135.1, 132.1, 129.4, 129.0, 126.6, 125.9, 123.6, 115.5, 113.7, 65.0, and 15.0; and LRMS (ESI–) *m/z* 324 (M–H)^−^.

Compound **6**: (*Z*)-5-(3-Hydroxy-4-methoxybenzylidene)-2-phenylthiazol-4(5*H*)-one.

Yield: 69%; ^1^H NMR (500 MHz, CDCl_3_) *δ* 8.20 (d, 2H, *J* = 8.0 Hz), 7.97 (s, 1H), 7.68 (t, 1H, *J* = 8.0 Hz), 7.55 (t, 2H, *J* = 8.0 Hz), 7.30 (d, 1H, *J* = 2.0 Hz), 7.21 (dd, 1H, *J* = 8.5, 2.0 Hz), 6.95 (d, 1H, *J* = 8.5 Hz), 5.83 (brs, 1H), and 3.97 (s, 3H); ^13^C NMR (100 MHz, CDCl_3_) *δ* 187.1, 183.7, 149.5, 146.4, 139.1, 135.2, 132.1, 129.4, 129.0, 127.5, 125.7, 124.4, 115.7, 111.1, and 56.4; and LRMS (ESI–) *m/z* 310 (M–H)^−^.

Compound **7**: (*Z*)-5-(2-Hydroxybenzylidene)-2-phenylthiazol-4(5*H*)-one.

Yield: 52%; ^1^H NMR (400 MHz, DMSO-*d*_6_) *δ* 10.74 (brs, 1H), 8.29 (s, 1H), 8.19 (d, 2H, *J* = 8.0 Hz), 7.79 (t, 1H, *J* = 8.0 Hz), 7.65 (t, 2H, *J* = 8.0 Hz), 7.62 (d, 1H, *J* = 8.4 Hz), 7.37 (t, 1H, *J* = 8.0 Hz), and 7.01–6.97 (m, 2H); ^13^C NMR (100 MHz, DMSO-*d*_6_) *δ* 187.1, 183.0, 158.9, 136.2, 134.1, 133.6, 131.9, 130.3, 129.5, 129.2, 125.2, 121.0, 120.6, and 117.1; and LRMS (ESI–) *m/z* 280 (M–H)^−^.

Compound **8**: (*Z*)-5-(4-Methoxybenzylidene)-2-phenylthiazol-4(5*H*)-one.

Yield: 78%; ^1^H NMR (400 MHz, DMSO-*d*_6_) *δ* 8.19 (d, 2H, *J* = 8.0 Hz), 8.00 (s, 1H), 7.82–7.77 (m, 3H), 7.66 (t, 2H, *J* = 8.0 Hz), 7.14 (d, 2H, *J* = 8.8 Hz), and 3.84 (s, 3H); ^13^C NMR (100 MHz, DMSO-*d*_6_) *δ* 186.6, 182.9, 162.6, 138.8, 136.1, 133.7, 131.9, 130.4, 129.1, 126.6, 123.7, 115.8, and 56.3; and LRMS (ESI+) *m/z* 296 (M+H)^+^.

Compound **9**: (*Z*)-5-(2,4-Dimethoxybenzylidene)-2-phenylthiazol-4(5*H*)-one.

Yield: 49%; ^1^H NMR (400 MHz, DMSO-*d*_6_) *δ* 8.17–8.14 (m, 3H), 7.77 (t, 1H, *J* = 8.0 Hz), 7.65–7.60 (m, 3H), 6.73 (d, 1H, *J* = 8.8 Hz), 6.68 (s, 1H), 3.92 (s, 3H), and 3.85 (s, 3H); ^13^C NMR (100 MHz, DMSO-*d*_6_) *δ* 186.4, 183.0, 164.8, 161.4, 135.9, 133.1, 132.0, 131.6, 130.3, 129.0, 123.2, 115.4, 107.8, 99.2, 56.8, and 56.5; and LRMS (ESI+) *m/z* 326 (M+H)^+^.

Compound **10**: (*Z*)-5-(3,4-Dimethoxybenzylidene)-2-phenylthiazol-4(5*H*)-one.

Yield: 43%; ^1^H NMR (500 MHz, CDCl_3_) *δ* 8.21 (d, 2H, *J* = 8.0 Hz), 8.01 (s, 1H), 7.68 (t, 1H, *J* = 8.0 Hz), 7.56 (t, 2H, *J* = 8.0 Hz), 7.32 (dd, 1H, *J* = 8.0, 2.0 Hz), 7.17 (d, 1H, *J* = 2.0 Hz), 6.98 (d, 1H, *J* = 8.0 Hz), 3.98 (s, 3H), and 3.96 (s, 3H); ^13^C NMR (100 MHz, CDCl_3_) *δ* 186.7, 183.6, 152.1, 149.7, 139.2, 135.2, 132.1, 129.4, 129.0, 127.0, 125.8, 124.0, 112.9, 111.6, 56.4, and 56.3; and LRMS (ESI+) *m/z* 326 (M+H)^+^, 348 (M+Na)^+^.

Compound **11**: (*Z*)-2-Phenyl-5-(3,4,5-trimethoxybenzylidene)thiazol-4(5H)-one.

Yield: 60%; ^1^H NMR (500 MHz, CDCl_3_) *δ* 8.20 (d, 2H, *J* = 8.0 Hz), 7.96 (s, 1H), 7.69 (t, 1H, *J* = 8.0 Hz), 7.56 (t, 2H, *J* = 8.0 Hz), 6.70 (s, 2H), 3.95 (s, 6H), and 3.93 (s, 3H); ^13^C NMR (100 MHz, CDCl_3_) *δ* 187.0, 183.4, 153.8, 141.2, 139.0, 135.4, 132.0, 129.5, 129.4, 129.1, 125.5, 108.2, 61.3, and 56.5; and LRMS (ESI+) *m/z* 356 (M+H)^+^.

Compound **12**: (*Z*)-5-(4-Hydroxy-3,5-dimethoxybenzylidene)-2-phenylthiazol-4(5H)-one.

Yield: 61%; ^1^H NMR (400 MHz, DMSO-*d*_6_) *δ* 9.60 (brs, 1H), 8.21 (d, 2H, *J* = 7.6 Hz), 7.95 (s, 1H), 7.78 (t, 1H, *J* = 7.6 Hz), 7.65 (t, 2H, *J* = 7.6 Hz), 7.10 (s, 2H), and 3.85 (s, 6H); ^13^C NMR (100 MHz, DMSO-*d*_6_) *δ* 186.3, 182.9, 149.0, 140.6, 139.9, 136.0, 131.9, 130.3, 129.1, 124.3, 123.0, 109.5, and 56.8; and LRMS (ESI–) *m/z* 340 (M–H)^−^.

Compound **13**: (*Z*)-5-(3-Bromo-4-hydroxybenzylidene)-2-phenylthiazol-4(5H)-one.

Yield: 72%; ^1^H NMR (500 MHz, DMSO-*d*_6_) *δ* 11.35 (brs, 1H), 8.19 (d, 2H, *J* = 8.0 Hz), 7.96 (s, 1H), 7.93 (s, 1H), 7.76 (t, 1H, *J* = 8.0 Hz), 7.67 (t, 2H, *J* = 8.0 Hz), 7.65 (d, 1H, *J* = 8.5 Hz), and 7.12 (d, 1H, *J* = 8.5 Hz); ^13^C NMR (100 MHz, DMSO-*d*_6_) *δ* 186.5, 182.8, 157.9, 137.7, 136.6, 136.2, 132.3, 131.8, 130.3, 129.2, 126.7, 124.1, 117.7, and 111.1; and LRMS (ESI–) *m/z* 358 (M–H)^−^, 360 (M+2–H)^−^.

Compound **14**: (*Z*)-5-(3,5-Dibromo-4-hydroxybenzylidene)-2-phenylthiazol-4(5H)-one.

Yield: 41%; ^1^H NMR (400 MHz, DMSO-*d*_6_) *δ* 8.21 (d, 2H, *J* = 8.0 Hz), 7.95 (s, 2H), 7.90 (s, 1H), 7.79 (t, 1H, *J* = 8.0 Hz), and 7.64 (t, 2H, *J* = 8.0 Hz); ^13^C NMR (100 MHz, DMSO-*d*_6_) *δ* 186.3, 182.9, 154.2, 136.4, 136.0, 135.1, 131.7, 130.3, 129.4, 128.4, 125.9, and 113.1; and LRMS (ESI–) *m/z* 436 (M–H)^−^, 438 (M+2–H)^−^, 440 (M+4–H)^−^.

#### 3.2.3. Procedure for the Synthesis of Compound **16** [43]

Bromoacetic acid (1.0 mL, 14.58 mmol) was added to a stirred solution of thiobenzamide (2.0 g, 14.58 mmol) in anhydrous ethyl acetate (28 mL). The reaction mixture was refluxed overnight. After cooling, the precipitate was filtered and washed with diethyl ether to produce 2-phenylthiazol-4(5*H*)-one hydrobromide (**15**, 2.60 g, and 69%) as a solid. Without further purification, **15** was used for the next step. A solution of **15** (2.60 g, 10.07 mmol) in pyridine (7 mL) was stirred for 1 h, then water (35 mL) was added and stirred for 20 min. The precipitate was filtered and washed with water to obtain 2-phenylthiazol-4(5*H*)-one (**16**, 1.648 g, and 92%) as a solid.

### 3.3. Kinetic and In Silico Studies and In Vitro Assays

#### 3.3.1. Inhibition Assay against Mushroom Tyrosinase

Minor modifications were made to the previous description of the inhibitory activity of mushroom tyrosinase [39]. Briefly, a mushroom tyrosinase aqueous solution (20 µL, 20 units) was added to a 96-well microplate containing a 170 µL substrate mixture of 345 µM of l-tyrosine solution and 17.2 mM phosphate buffer (pH 6.5), as well as 10 µL of (*Z*)-BPT derivatives’ solution at various concentrations depending on their tyrosinase inhibitory activity or 10 µL kojic acid solution at various concentrations (0, 25, 50, and 100 μM). After 30 min incubation at 37 °C, the amount of dopachrome produced in the assay mixture was measured at 492 nm using a microplate reader (VersaMax^TM^, Molecular Devices, Sunnyvale, CA, USA) for determining the remaining tyrosinase activity. To obtain the IC_50_ values for each test sample, dose-dependent inhibition studies were performed in triplicates using three to five different concentrations of each test sample. The concentration at which there was 50% inhibition along the Y-axis was used to calculate the IC_50_ value.

#### 3.3.2. Kinetic Studies on the Inhibition of Mushroom Tyrosinase by (Z)-BPT Derivatives **1**–**3**

To determine the mode of action, Lineweaver–Burk plots of (*Z*)-BPT derivatives **1**–**3** were acquired. In brief, 10 µL of (*Z*)-BPT derivatives **1**–**3** (final concentrations:0, 3, 6, or 12 µM for **1**; 0, 0.05, 0.1, or 0.2 µM for **2**; 0, 2.5, 5, or 10 µM for **3**) was added to a 96-well plate containing a substrate mixture (170 µL) consisting of an aqueous solution of L-dopa at final concentrations of 0.5, 1.0, 2.0, or 4.0 mM and 14.7 mM potassium phosphate buffer (pH 6.5), and a mushroom tyrosinase aqueous solution (20 µL, 20 units). Using a microplate reader (VersaMax^TM^, Molecular Devices, Sunnyvale, CA, USA) to measure the increase in absorbance at a wavelength of 492 nm (ΔOD_492_/min), the initial rate of dopachrome synthesis in the reaction mixture was estimated. Using five different concentrations of l-dopa, Lineweaver–Burk plots (inverse of the response rate (1/V) vs. inverse of l-dopa concentration (1/(S)) were created. The convergence points of the four plot lines were used to define the mode of action of tyrosinase inhibition.

#### 3.3.3. In Silico Study of Kojic Acid and (Z)-BPT Derivatives **1**–**3** on Mushroom Tyrosinase (mTYR)

The Schrödinger Suite (2021-1) was used for in silico analysis in accordance with the previously described procedures [35] with a few minor modifications. The Protein Preparation Wizard in Maestro12.4 was provided with the crystal structure of *m*TYR (PDB:2Y9X, *Agaricus bisporus*). The structure was then optimized by including hydrogen atoms, removing water molecules that were more than 3 Å away from the enzyme, and minimizing the structure. The binding site of tropolone was used to designate the glide grid of the enzyme. The structures of kojic acid and its derivatives **1**–**3** were imported into the CDXML format to the entry list of Maestro. The structures of kojic acid and its derivatives **1**–**3** were generated using LigPrep prior to ligand docking. Using glide from the task list [57], the ligand structures were docked to the glide grid of the enzyme. The glide extra precision (XP) approach was used to determine the binding affinity and interactions between ligands and proteins [58].

#### 3.3.4. In Silico Study of Kojic Acid and (Z)-BPT Derivatives **1**–**3** on Human Tyrosinase Homology Model

To perform in silico docking simulations, the *h*TYR homology model previously created using the Swiss-Model online server and the Schrödinger Suite (2020-2) was used [34]. Kojic acid and (*Z*)-BPT derivatives **1**–**3** were docked using the *h*TYR homology model using the same procedures as those described above for *m*TYR docking.

#### 3.3.5. B16F10 Murine Melanoma Cell Culture

B16F10 cells were purchased from the American Type Culture Collection (ATCC, Manassas, VA, USA). Penicillin, trypsin, streptomycin, phosphate-buffered solution (PBS), fetal bovine serum (FBS), and Dulbecco’s modified Eagle’s medium (DMEM) were purchased from Gibco/Thermo Fisher Scientific (Waltham, MA, USA). A DMEM solution containing penicillin–streptomycin solution (10,000 U/mL) and 10% heat-inactivated FBS was used to cultivate B16F10 cells at 37 °C in a humid environment with 5% CO_2_. These cells were cultivated in a 6- or 96-well culture plate, and cell viability, anti-tyrosinase activity, and anti-melanogenesis activity experiments were conducted.

#### 3.3.6. Cytotoxicity Analysis of (Z)-BPT Derivatives **1**–**3** in B16F10 Melanoma Cells

Cytotoxicity studies of (Z)-BPT derivatives **1**–**3** were performed on B16F10 melanoma cells using the previously established EZ-Cytox (EZ-3000, DoGenBio, Seoul, Republic of Korea) test [33], and cytotoxicity studies of (*Z*)-BPT derivatives **1**–**3** were conducted on B16F10 melanoma cells. Briefly, B16F10 cells were seeded in a 96-well plate at a density of 1 × 10^4^ cells/well and cultivated at 37 °C for 24 h in a humid environment with 5% CO_2_. Next day, the cells were treated with kojic acid or (*Z*)-BPT derivatives **1**–**3** at six different concentrations (0, 1, 2, 5, 10, and 20 µM) and incubated at 37 °C for 48 and 72 h, respectively, in a humid environment with 5% CO_2_. The cells were then incubated at 37 °C for 2 h after 10 µL of the EZ-Cytox solution had been added to each well. To determine the cell viability, the absorbance of each well was measured at 450 nm using a microplate reader (VersaMax^TM^, Molecular Devices, Sunnyvale, CA, USA). Each assay was performed in triplicates.

#### 3.3.7. Anti-Melanogenesis Assay of Kojic Acid and (Z)-BPT Derivatives **1** and **2** in B16F10 Cells

The melanin content assay standard procedure [37,59] was used to investigate the anti-melanogenic effect of (*Z*)-BPT derivatives **1** and **2** with a few minor modifications. Briefly, B16F10 melanoma cells were seeded in each well of a 6-well plate at a density of 1 × 10^5^ cells/well and allowed to adhere to the well bottom under conditions that were identical to those used for cell culture. After 24 h, the cultured B16F10 cells were treated with kojic acid (20 µM) or four different concentrations (0, 5, 10, or 20 µM) of (*Z*)-BPT derivatives **1** and **2** for 1 h before stimulation with 200 µM IBMX and 1 µM α-MSH. For 72 h, B16F10 cells were incubated at 37 °C in a humid environment containing 5% CO_2_ after being treated with IBMX and α-MSH. To determine the intracellular melanin content, B16F10 cells stimulated with IBMX and α-MSH for 72 h were rinsed twice with PBS and incubated in 200 µL 1N-NaOH solution including 10% dimethyl sulfoxide (DMSO) at 60 °C. The cell lysates were transferred to a 96-well plate after an hour of incubation, and the absorbance of the melanin content was calculated using a VersaMax^TM^ microplate reader (Molecular Devices, Sunnyvale, CA, USA) to determine intracellular melanin levels. Each experiment was performed independently in triplicates.

#### 3.3.8. Anti-Tyrosinase Activity Assay of Kojic Acid and (Z)-BPT Derivatives **1** and **2** in B16F10 Cells

The cellular tyrosinase activities of kojic acid and (*Z*)-BPT derivatives **1** and **2** were investigated using an anti-tyrosinase activity assay based on the oxidation rate of l-DOPA previously reported [60]. Briefly, B16F10 melanoma cells were seeded in a 6-well plate at a density of 1 × 10^5^ cells/well and allowed to adhere to the bottom of each well under identical conditions to the cell culture. B16F10 cells were incubated for 24 h before treatment with kojic acid (20 µM) or various concentrations (0, 5, 10, or 20 µM) of (*Z*)-BPT derivatives **1** and **2**. After incubation for 1 h, the tyrosinase activity of the cells was increased by treatment with 200 µM IBMX and 1 µM α-MSH. The cells were incubated for 72 h under identical conditions to cell culture before being rinsed twice with PBS, exposed to a 100 µL lysis buffer solution consisting of 90 µL phosphate buffer (50 mM) at pH 6.5, 5 µL PMSF (2 mM), and 5 µL Triton X-100 (20%), and then incubated for 30 min at −80 °C. The cell lysates were placed into microcentrifuge tubes after defrosting and they were centrifuged at 12,000 rpm at 4 °C for 30 min. The supernatants (80 µL) of the lysates were combined with 20 µL of 10 mM l-DOPA in a 96-well plate. Using a microplate reader used for the melanin content assay, the mixture’s absorbances were measured every 10 min for 1 h at 37 °C and 492 nm. Each experiment was performed independently in triplicates.

#### 3.3.9. ABTS Cation-Free Radical Scavenging Assay of (Z)-BPT Derivatives **1**–**14**

The antioxidant properties of (*Z*)-BPT derivatives **1**–**14** were assessed using the ABTS cation-free radical scavenging assay, as previously described [61]. Potassium persulfate solution (10 mL, 2.45 mM) in distilled water was mixed with the ABTS solution (10 mL, 7 mM) in distilled water, and the mixture was incubated for 16 h at room temperature in the dark to produce ABTS cation-free radicals. Then, methanol was used to dilute the ABTS cation-free radical solution to an absorbance of 0.70 ± 0.02 at 734 nm. The test samples (10 µL; 90% EtOH + 10% DMSO) with a final concentration of 100 µM were added to the diluted ABTS cation-free radical solution (90 µL) and incubated in the dark at room temperature for 2 min. The absorbance of each well was measured at 734 nm using a VersaMax ™ microplate reader (Molecular Devices, Sunnyvale, CA, USA). The ABTS cation-free radical scavenging ability of the (*Z*)-BPT derivatives was determined using the following formula:ABTS cation radical scavenging activity (%) = (A_C_ − A_T_) × 100/A_C_(1)
where A_C_ is the absorbance of the untreated control, and A_T_ is the absorbance of the tested compounds.

#### 3.3.10. DPPH Radical Scavenging Assay of (Z)-BPT Derivatives **1**–**14**

The antioxidant properties of (*Z*)-BPT derivatives **1**–**14** against DPPH (2,2-diphenyl-1-picrylhydrazyl) radicals were examined using a DPPH radical scavenging assay, as previously described with minor modifications [62]. DPPH solution (180 µL, 0.2 mM) in methanol was mixed with a (*Z*)-BPT derivative solution (**1**–**14**, 20 µL, 10 mM) in dimethyl sulfoxide in each well of a 96-well plate, and the mixture was kept for 30 min in the dark at room temperature. The absorbance at 517 nm was measured using a VersaMax ™ microplate reader (Molecular Devices, Sunnyvale, CA, USA). The capacity of the derivatives to scavenge DPPH radicals was compared with that of l-ascorbic acid, which was used as the positive control. Each experiment was run independently in triplicates.

#### 3.3.11. Intracellular ROS Scavenging Activity Assay of (Z)-BPT Derivatives **1**–**14**

The intracellular ROS scavenging activity was assessed as described by Lebel and Bondy [54] and Ali et al. [55]. The assay was evaluated using DCFH-DA as the ROS-sensitive fluorescence probe. In brief, B16F10 melanoma cells at a density of 1 × 10^4^ cells/well were incubated for 24 h in 96-well black plates and treated with 20 µM (*Z*)-BPT derivatives **1**–**14** for 2 h. Next, 10 µL of SIN-1 (20 µM in 50 mM sodium phosphate buffer, pH 7.4) was added for 1 h to stimulate B16F10 cells and induce ROS production. The cells were incubated with 20 µM DCFH-DA at 37 °C for 30 min. Fluorescence was measured every 5 min for 30 min using a microplate reader (Berthold Technologies GmbH & Co., Vienna, Austria) at 485 nm (excitation wavelength) and 535 nm (emission wavelength).

#### 3.3.12. In Vitro ROS Scavenging Activity Assay of (Z)-BPT Derivatives **1**–**14**

The ROS scavenging abilities of (*Z*)-BPT derivatives **1**–**14** and Trolox were assessed in accordance with methods previously reported by Lebel and Bondy [54]. Briefly, 50 mM phosphate buffer (pH 7.4) was dissolved with esterase (6 units/mL) and 1.25 mM DCFH-DA (2′,7′-dichlorodihydrofluorescein diacetate) from Molecular Probes (Eugene, OR, USA), maintained at 37 °C for 30 min, and kept on ice in the dark until use. (*Z*)-BPT derivatives (final concentration: 40 µM; 10 µL) and SIN-1 (3-morpholinosydnonimine) (10 µM; 10 µL), dissolved in DMSO and phosphate buffer (180 µL), respectively, were added to each well of a black 96-well plate and kept in the dark for 30 min. The esterase-DCFH-DA mixture (50 µL) was then added to each well. Using a microplate reader (Berthold Advances GmbH & Co., Bad Wildbad, Germany), the fluorescence intensities of 2′,7′-dichlorofluorescein (DCF or oxidized DCFH) were examined at excitation and emission wavelengths of 485 nm and 535 nm, respectively. Trolox was used as the reference control.

#### 3.3.13. Peroxynitrite (ONOO^−^) Scavenging Assay of (Z)-BPT Derivatives **1**–**14**

The peroxynitrite scavenging activities were determined as described by Kooy et al. [63], with a few modifications. This technique measures the fluorescence of rhodamine 123, which is produced quickly when non-fluorescent DHR 123 (dihydrorhodamine 123) (Molecular Probes, Eugene, OR, USA) is used. Potassium chloride (5 mM), sodium chloride (90 mM), sodium phosphate (50 mM), and diethylenetriamine pentaacetic acid (DTPA, 100 µM) were used to prepare the pH 7.4 rhodamine buffer solution. DHR 123 at a final concentration of 5 µM was used. The buffer solution was prepared in advance for use in this test and was kept on ice. (*Z*)-BPT derivatives (10 µL), with a final concentration of 50 µM, were dissolved in DMSO and combined with SIN-1 (10 µL, 50 µM) and rhodamine buffer solution (180 µL) in each well of a black 96-well plate. The fluorescence intensities of the generated rhodamine 123 were measured using a fluorescent plate reader (Berthold Advances GmbH & Co., Bad Wildbad, Germany) at excitation and emission wavelengths of 485 and 535 nm, respectively. After deducting the background fluorescence from the final fluorescence intensity, the peroxynitrite scavenging activity was determined. l-penicillamine was used as a positive control, and the results are shown as the mean ± standard deviation.

#### 3.3.14. Western Blot Assay of Tyrosinase Protein

After stimulating murine melanoma (B16F10) cells with α-MSH and IBMX for 24 h in the presence or absence of the tested derivatives at the indicated concentrations, the cells were lysed with lysis buffer containing protease and phosphatase inhibitors [38]. Protein concentrations were determined using a BCA protein assay kit, according to the manufacturer’s instructions (Pierce, Rockford, IL, USA). Lysate proteins in each sample (10–20 µg) were subjected to 9% sodium dodecyl sulfate–polyacrylamide gel electrophoresis and electrophoretically transferred to a PVDF membrane (Millipore, Billerica, MA, USA) using a semi-dry system (Bio-Rad, Hercules, CA, USA). The membranes were incubated with 5% non-fat milk in TBST (0.1 M Tris-HCl, pH 7.5, 1.5 M NaCl, and 1% Tween 20) for 2 h. After blocking, the membranes were incubated with primary antibodies at 4 °C overnight. The following antibodies were used: tyrosinase and β-actin primary antibodies, as well as horseradish peroxidase-conjugated anti-mouse and anti-goat secondary antibodies (Santa Cruz Biotechnology, Dallas, TX, USA). The protein bands were visualized using the SuperSignal West Pico Chemiluminescence assay kit (Advansta, San Jose, CA, USA) and the Davinch–Chem program (Davinch-K, Seoul, Republic of Korea). The relative intensities were acquired using CS Analyzer 3.2 (Densitograph) image analysis software (http://www.attokorea.co.kr, accessed on 21 November 2022).

#### 3.3.15. Statistical Analysis

The significance of the differences between treatment groups was determined using one-way analysis of variance (ANOVA) and the Bonferroni post hoc test. The statistical analysis was performed using GraphPad Prism 5 (La Jolla, CA, USA). All results are presented as the mean ± standard error of the mean (SEM). *P*-values with two sides less than 0.05 were regarded as statistically significant.

## 4. Conclusions

Novel anti-melanogenic compound (*Z*)-5-benzylidene-2-phenylthiazol-4(5*H*)-one ((*Z*)-BPT) derivatives were designed and synthesized from thiobenzamide and bromoacetic acid. Three (*Z*)-BPT derivatives (**1**–**3**) inhibited mushroom tyrosinase more effectively than kojic acid. Using kinetic studies, their modes of inhibition were demonstrated: **1** and **2** were competitive inhibitors, and **3** was a mixed-type inhibitor; the results were supported by docking simulation. Experiments using B16F10 cells revealed that derivatives **1** and **2** decreased the melanin production by inhibiting the tyrosinase activity and suppressing tyrosinase protein expression. Several (*Z*)-BPT derivatives exert strong antioxidant activities, scavenging DPPH and ABTS cation radicals, peroxynitrite, and intracellular and in vitro ROS.

## Data Availability

Not applicable.

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
