# Peer review of "Design, Synthesis, In Vitro, and In Silico Insights of 5-(Substituted benzylidene)-2-phenylthiazol-4(5H)-one Derivatives: A Novel Class of Anti-Melanogenic Compounds"

_molecules, 2023, doi:10.3390/molecules28083293_

Round 1

Reviewer 1 Report

The manuscript “Design, Synthesis, In Vitro, and In Silico Insights of 5-(Substituted benzylidene)-2-phenylthiazol-4(5H)-one Derivatives: A Novel Class of Anti-melanogenic Compounds” describes the synthesis, tyrosinase inhibitory, anti-melanogenic, and antioxidant activities of title compounds. The results of biological evaluation are promising, the manuscript is generally well-written and I think it is suitable for publication in Molecules after some revision (please, see the list below). 

1) Z-configuration of a double C=C bond was assigned elegantly based on vicinal J(C-H) value, but is was measured for one compound and it is still indirect method, J value depends on many factors and can vary for different trisubstituted C=C double bonds. Some additional confirmation will not be redundant. It can be single crystal XRD for some of the products or DFT calculation showing that Z-configuration is more thermodynamically favorable and thus more plausible.  

2) Scheme 1, reaction under conditions b – acid-base interaction is usually instant, why 1 h reaction time is needed?

3) Scheme 1, reactions under conditions b and c – bases are used on both stages, why this two stages can not go one-pot? For example, with only piperidine, but with increased quantity to neutralize HBr.

4) The quantity of self-citations should be reduced, if possible.

Author Response

The manuscript “Design, Synthesis, In Vitro, and In Silico Insights of 5-(Substituted benzylidene)-2-phenylthiazol-4(5H)-one Derivatives: A Novel Class of Anti-melanogenic Compounds” describes the synthesis, tyrosinase inhibitory, anti-melanogenic, and antioxidant activities of title compounds. The results of biological evaluation are promising, the manuscript is generally well-written and I think it is suitable for publication in Molecules after some revision (please, see the list below). 

1) Z-configuration of a double C=C bond was assigned elegantly based on vicinal J(C-H) value, but is was measured for one compound and it is still indirect method, J value depends on many factors and can vary for different trisubstituted C=C double bonds. Some additional confirmation will not be redundant. It can be single crystal XRD for some of the products or DFT calculation showing that Z-configuration is more thermodynamically favorable and thus more plausible.  

Thank you for your valuable comment. We investigated whether a DFT experiment is possible. Unfortunately, we do not have a program that can perform DFT experiments, and we do not have experts who can perform this program. Please understand. Therefore, we searched for known compounds among our final compounds. Since only compound 8 was a known compound, 1H and 13C NMR spectra of the reported compound (having (Z)-configuration) were compared with those of compound 8, and it was confirmed that they were identical to each other. The spectra of the reported compound and compound 8 were added to the Supplementary Information (S44 – S47). In addition, it was confirmed that only (Z)-isomers were obtained in all similar reactions. See references below.

(1) Organic Letters (2020), 22(17), 6868-6872; (2) Advanced Synthesis & Catalysis (2020), 362(5), 1058-1063; (3) Journal of Medicinal Chemistry (2011), 54(6), 1943-1947.

2) Scheme 1, reaction under conditions b – acid-base interaction is usually instant, why 1 h reaction time is needed?

Thank you for your valuable comment. As the reviewers have argued, acid-base reactions generally occur rapidly. However, since we have experienced cases where this is not the case (when acidic salts of amines are made to improve the solubility of some pharmaceuticals in water), sufficient reaction time (1 hour) was provided for the acid-base reaction to be sufficiently completed.

3) Scheme 1, reactions under conditions b and c – bases are used on both stages, why this two stages can not go one-pot? For example, with only piperidine, but with increased quantity to neutralize HBr.

Thank you for your creative suggestion. At that time, we weren’t aware of what the reviewer suggests. I think that one-pot reaction including conditions b and c is possible. Thank you again for your valuable suggestion.

4) The quantity of self-citations should be reduced, if possible.

Thank you for your valuable suggestion. We have deleted 5 self-citations, as suggested by the reviewer.

Reviewer 2 Report

The manuscript describes the design and synthesis of (Z)-5-Benzylidene-2-phenylthiazol-4(5H)-one derivatives, their characterisation (1H and 13C NMR, LRMS) and evaluation of their biological activity. The aim of this study was to assess their anti-melanogenic potential. With three of these new compounds that showed the lowest IC50 values against mushroom tyrosinase, studies to determine inhibition type and in silico docking were also conducted. Furthermore, in vitro cell viability was tested on B16F10 cells, and due to toxicity of the compound 3 in these cells, only derivatives 1 and 2 were further tested on melanin production and B16F10 cellular tyrosinase activity and expression. All 14 derivatives were tested for antioxidant capacities with ABTS, DPPH, ROS (by two assays – for intracellular and in vitro ROS) and peroxynitrite.

The manuscript is well written, properly organized and easy to read. The introduction refers to relevant previous research, and the experiments are described in the materials and methods section in sufficient details, also referring to previous studies, conducted mainly by the same research group. The results are clearly presented and explained, with appropriate discussion and conclusions.

Minor corrections to be made:

I suggest using different signs for used benzaldehydes (capital letters A – N or Roman numerals I -XIV instead of ‘a – n’) in Scheme 1. not to be confused with letters referring to reaction conditions (a - c).

Line 205. In Figure 2. caption – it should be added what are the five different concentration values of L-DOPA substrates that were used in the experiments.

Line 232. In Figure 3. concentrations of L-DOPA (0.25, 0.5, 1 and 2 mM) are not the same as in the figure caption (0.5, 1, 2, and 4 mM).

Section 2.10. all the text is in bold.

Line 497. Figure 13. – check p-values (statistical significance), on the graphs and in the Figure caption.

Author Response

The manuscript describes the design and synthesis of (Z)-5-Benzylidene-2-phenylthiazol-4(5H)-one derivatives, their characterisation (1H and 13C NMR, LRMS) and evaluation of their biological activity. The aim of this study was to assess their anti-melanogenic potential. With three of these new compounds that showed the lowest IC50 values against mushroom tyrosinase, studies to determine inhibition type and in silico docking were also conducted. Furthermore, in vitro cell viability was tested on B16F10 cells, and due to toxicity of the compound 3 in these cells, only derivatives 1 and 2 were further tested on melanin production and B16F10 cellular tyrosinase activity and expression. All 14 derivatives were tested for antioxidant capacities with ABTS, DPPH, ROS (by two assays – for intracellular and in vitro ROS) and peroxynitrite.

The manuscript is well written, properly organized and easy to read. The introduction refers to relevant previous research, and the experiments are described in the materials and methods section in sufficient details, also referring to previous studies, conducted mainly by the same research group. The results are clearly presented and explained, with appropriate discussion and conclusions.

Minor corrections to be made:

I suggest using different signs for used benzaldehydes (capital letters A – N or Roman numerals I -XIV instead of ‘a – n’) in Scheme 1. not to be confused with letters referring to reaction conditions (a - c).

Thank you for your valuable suggest. As suggested by the reviewer, we have used different signs for used benzaldehydes, capital letters A – N in Scheme 1.

Line 205. In Figure 2. caption – it should be added what are the five different concentration values of L-DOPA substrates that were used in the experiments.

Thank you for your valuable comment. As suggested by the reviewer, we have added the values of the five different concentrations of the L-DOPA substrate in figure caption.

Line 232. In Figure 3. concentrations of L-DOPA (0.25, 0.5, 1 and 2 mM) are not the same as in the figure caption (0.5, 1, 2, and 4 mM).

Thank you for your comment. In Figure 3, mistakes about L-DOPA concentrations have been corrected. Concentrations of 0.25, 0.5, 1, and 2 mM were modified to 0.5, 1, 2, and 4 mM.

Section 2.10. all the text is in bold.

Thank you for your comment. We have revised the text style of ‘Section 2.10’ to a plain style.

Line 497. Figure 13. – check p-values (statistical significance), on the graphs and in the Figure caption.

Thank you for your detailed check. We have corrected the statistical significance error in the Figure caption.

Round 2

Reviewer 1 Report

The Authors have improved the manuscript as it was possible. I think it can be accepted for publication.